# The Effect of Transcranial Direct Current Stimulation on Error Rates in the Distractor-Induced Deafness Paradigm

**DOI:** 10.3390/brainsci12060738

**Published:** 2022-06-04

**Authors:** Lars Michael, Ana Böke, Henry Ipczynski

**Affiliations:** Department of Psychology, Medical School Berlin, 14197 Berlin, Germany; ana.boeke@student.medicalschool-berlin.de (A.B.); henry.ipczynski@student.medicalschool-berlin.de (H.I.)

**Keywords:** transcranial direct current stimulation (tDCS), distractor-induced deafness, attention, conscious perception

## Abstract

To further understand how consciousness emerges, certain paradigms inducing distractor-induced perceptual impairments are promising. Neuro-computational models explain the inhibition of conscious perception of targets with suppression of distractor information when the target and distractor share the same features. Because these gating mechanisms are controlled by the prefrontal cortex, transcranial direct current stimulation of this specific region is expected to alter distractor-induced effects depending on the presence and number of distractors. To this end, participants were asked to perform an auditory variant of the distractor-induced blindness paradigm under frontal transcranial direct current stimulation (tDCS). Results show the expected distractor-induced deafness effects in a reduction of target detection depending on the number of distractors. While tDCS had no significant effects on target detection per se, error rates due to missed cues are increased under stimulation. Thus, while our variant led to successful replication of behavioral deafness effects, the results under tDCS stimulation indicate that the chosen paradigm may have difficulty too low to respond to stimulation. That the error rates nevertheless led to a tDCS effect may be due to the divided attention between the visual cue and the auditory target.

## 1. Introduction

An integral part of the perception of ourselves is the conscious experience of our environment, but for the occurrence and the intensity of this conscious experience, some preconditions must be fulfilled. For example, stimuli usually enter consciousness particularly vividly when they are the focus of attention. However, the conscious impression can also be significantly weakened or even absent if stimuli are ignored.

An experimental setup that is well suited for investigating the interplay between attending and ignoring stimuli is the distractor-induced blindness/deafness paradigm. Here, the detection of visual or auditory features is modulated by attentional mechanisms. In a typical arrangement, an easily detectable, centrally presented stimulus serves as a cue. For the study of distractor-induced blindness, this central area is surrounded by a random walk by a set of white (or colored) dots (or bars) on a gray background, which, depending on the feature examined, show episodes of coherent motion (e.g., [1]), changes in orientation (e.g., [2]) or changes in color (e.g., [3]). The subject’s task is to attend to the cue to detect the simultaneously presented visual feature in the surroundings as a target. Presentations of the same visual feature prior to the cue serve as distractors and shall be ignored. In such an arrangement, the detection performance depends on the presence and number of distractors: with an increasing number of distractors, distractor-induced blindness is expressed more, whereas the effect is absent in trials without distractors (e.g., [4]). In a variant of this arrangement, the distractor–induced deafness, comparable results were obtained when specific frequencies in an auditory stimulation served as distractors and targets [5].

This feature-specific effect supports the notion that these distractor-induced processes rely on top-down mechanisms that control attentional target selection when target-defining features are specified in advance. Distractors lead to a negative attentional set (or the inhibition of the task set), which suppresses conscious target perception. This view is supported by event-related potentials recorded during such experiments, which revealed an accumulation of frontal negativity induced by distractors [3,5,6,7].

According to a neuro-computational model [8], the target’s access to consciousness is impeded when inadequate distractor-induced response tendencies are suppressed by a basal ganglia hyperdirect pathway, inhibiting the prefrontal cortex. If a target closely follows such a distractor, temporal aftereffects of distractor suppression prevent target identification. Coming from the prefrontal cortex, a direct inhibitory pathway to the basal ganglia also exists, which is able to remove the built-up inhibition. It is assumed that this direct pathway enables specific cognitive cortical representations. While this direct pathway is capable of gating information into consciousness, the inhibitory hyperdirect pathway suppresses the gating of irrelevant information into consciousness. When a target closely follows a distractor, the hyperdirect pathway’s suppression is not reduced fast enough for the target to receive access, resulting in distractor-induced blindness.

Transcranial direct current stimulation (tDCS) is a well-established method to investigate and alter brain physiology [9]. When the anode is placed over a certain area, it depolarizes neuron membranes of that region, which in turn need less additional activity to induce an action potential and are therefore easier to excite, whereas cathodes lead to hyperpolarized membranes, which are harder to excite [10].

Such effects on activity and excitability occur within seconds of stimulation and vanish when stimulation is stopped. Stimulation of cortical areas is often used for studying their functional specialization or its necessity in certain perceptual functions [9]. Studies have shown that tDCS can alter oscillatory cortical activity. For example, both anodal and cathodal tDCS with electrodes placed over the DLPFC were able to alter working memory performance [11], but it was also observed that there is strong heterogeneity in possible effects of tDCS and large inter- and intra-individual variability among test subjects, making it hard to hypothesize on certain effects [10,12].

The present study aims to further clarify the underlying mechanisms of distractor-induced perceptual impairments and the possibility of influencing distractor-induced effects via tDCS. To this end, the distractor-induced effects of auditory stimuli are replicated. In contrast to previous work on distractor-induced deafness [5,13], the present study uses a setup in which temporally distinct tones are presented as distractors and target stimuli. This is intended to provide greater comparability to the studies of DIB, in which highly salient visual events were presented in the global stream. Therefore, a (local) visual cue stimulus should also be presented in the present study so that a separation from the (auditory) distractor-target stream can still be maintained. We, therefore, expect a clear reduction of detection rates for targets with an increasing number of distractors. Furthermore, the effects of tDCS on detection rates for targets as a function of the number of distractors are considered. Since visual cue stimuli and auditory target stimuli are presented simultaneously in the present study, attention to the cues is examined in addition to consideration of the detection of the target stimuli.

## 2. Materials and Methods

### 2.1. Participants

A minimum sample size of 19 participants (for f = 0.25, α = 0.01; 1 − β = 0.80) was calculated in advance using G*Power [14]. The final sample consisted of 22 participants (13 females and 9 males) with an age range of 19–59 years (M = 24.13 years, SD = 8.53 years). Four additional participants had to be excluded from further analyses due to high error rates (more than 30% false alarms) in control conditions where no targets or no cues were presented. All participants had normal or corrected-to-normal vision and hearing and fulfilled common inclusion criteria for tDCS studies [15]. All subjects were recruited by advertisement and received course credit after giving informed consent. The study was conducted in accordance with the Declaration of Helsinki and approved by the Institutional Ethics Committee of the Medical School Berlin (MSB-2022/89, 4 April 2022).

### 2.2. Distractor-Induced Deafness

In previous studies that focused on distractor-induced deafness [5,13], the target was defined by a transient increase in amplitude in a continuous sinusoidal tone, which was to be detected if accompanied or preceded by a deviant tone (the cue). Both events were embedded in separate streams in a binaural rapid serial auditory presentation.

The arrangement in the present study is more similar to arrangements used in several studies examining distractor-induced blindness (e.g., [2,16]), whereas instead of a global visual stream, an auditory stimulation was used.

The (local) visual stream in which the cue is presented consists of an orientation change of a central fixation bar every 150 ms (consisting of three white squares with 0.5° × 0.5°). The used orientation angles were −45°, −22.5°, 22.5°, and 45°. The white fixation bar was centered in a black square (3.0° × 3.0°) and surrounded by a gray background screen RGB (100 100 100).

The auditory stream in which context tones, distractors, and targets were presented consisted of sinus tones with a length of 50 ms and an ISI of 100 ms. The auditory stimuli were synchronized with the visual stream, leading to a sound onset with every onset of an orientation change. The frequencies of the context tones ranged from 500 to 600 Hz; the specific frequencies were 500, 520, 540, 560, 580, and 600 Hz. Distractors and targets were presented with 350 Hz.

Each trial was started by a button press. Participants were instructed to keep fixation on the central bar throughout the 5250 ms of a trial. At the moment of cue detection (a white cross presented for 150 ms instead of a bar) in the visual stream, participants had to switch attention to the auditory stream to detect the target (the tone with 350 Hz). Tones with 350 Hz presented prior to the cue had to be ignored and served as distractors

The cue was presented between 2250 and 3750 ms after the beginning of the trial. In order to avoid temporal uncertainty, distractors were not presented in an interval of 600 ms prior to the cue. The final distractor was always presented 600 to 1050 ms before cue onset. After each trial, subjects had to indicate if the target was detected or not or if the cue was not presented by pressing a corresponding button.

For each participant, one block of trials was presented comprising 150 trials, leading to an overall duration of approximately 20 min. In three experimental conditions, either zero, three, or six distractors were presented (30 trials each). In two control conditions, 30 additional trials without a target and another 30 trials without a cue were presented. All trials were presented in randomized order.

### 2.3. tDCS

All participants went through three tDCS conditions, i.e., two tDCS and one sham stimulation, each administered in a separate session. The order of the conditions was counterbalanced. The three sessions were performed around the same time of day with a minimum of 48 h between them.

Stimulation was delivered by the DC-Stimulator plus (neuroConn GmbH, Illmenau, Germany). The 5 cm × 7 cm conductive rubber electrodes were covered in saline-soaked sponge pockets and fixated with an elastic band at positions F7 and F8 of the international 10–20 system [17]. For left anodal stimulation, the anode was placed on F7 and the cathode on F8 and for right anodal stimulation, vice versa. The placement of electrodes was chosen as such because, in this position, frontal brain areas are stimulated well without spreading too much to other cortical areas [18,19]. We preferred this montage primarily to keep more posterior areas out of stimulation since the neuro-computational model [8] also includes the basal ganglia, thalamus, and motor areas. Additionally, the stimulated regions are known for playing a role in response selection and inhibition processes [18].

The current intensity was 1 mA, applied constantly. There is evidence that in arrangements requiring different cognitive loads, there are no differences between the effects of stimulation at 1 mA and 2 mA [20]. This appears comparable to the requirements in the present study, where different numbers of distractors must be ignored in different conditions. The stimulation duration was set to 20 min with an additional fade-in/fade-out period of 30 s, resulting in an overall stimulation duration of 21 min.

In the sham condition, electrode positioning was kept the same. The fade-in period was followed immediately by the fade-out period to give the impression that an actual stimulation was started.

### 2.4. Procedure

In the first session, participants were informed of the details of the experiment and signed a declaration of consent.

Participants were seated in front of a computer screen inside a sound-attenuated and constant-lit chamber. The viewing distance was maintained at approximately 60 cm. Stimuli were generated and controlled electronically using the E-Prime 2.0 software (Psychology Software Tools, Pittsburgh, PA, USA) and displayed on a 21-inch LCD monitor. Sounds were presented at a comfortable intensity of about 60 dB SPL from two loudspeakers (Logitech Z200, Logitech international S.A., Apples, Switzerland), positioned left and right from the monitor.

After a brief instruction for the distractor-induced deafness task, a short exercise phase of 10 trials was added before the test phase in order to make the participants better acquainted with the task. After the tDCS electrodes were put in position, the tDCS stimulation and the experiment on the computer were started simultaneously, both taking approximately 20 min.

### 2.5. Analytical Approach and Statistical Analysis

The data were analyzed in Jamovi (version 2.2.5.0 for Windows). First, error rates in the control condition were computed. Participants with a rate of more than 30% false alarms in at least one of the control conditions where no targets or no cues were presented were excluded from further analyses. This approach is based on previous studies on distractor-induced blindness [2,16], where these errors were seen as indicators of lack of task understanding, lack of attention, or random response behavior.

For each remaining participant, the mean detection rate and error rates for missed cues were computed separately for each experimental condition. The detection rate was determined as the percentage by which subjects reported perceiving the actual target stimuli presented in each condition. The error rate for missed cues was determined as the percentage by which subjects reported not perceiving a cue stimulus in each condition (even though cue stimuli were presented there in each trial). Data were statistically analyzed using a repeated-measures ANOVA with number of distractors (0, 3, and 6) and anodal tDCS condition (left PFC, right PFC, sham stimulation) as within-subject factors. Greenhouse–Geisser corrected *p*-values are reported to minimize the possible effects of sphericity violations.

## 3. Results

In control trials without a cue, the participants correctly indicated the cue’s absence (the mean rate of correct rejection in the three tDCS conditions were between 96.8% and 98.0%). In control trials without a target, the mean rates of correct rejection ranged between 94.7% and 95.6%.

### 3.1. Detection Rates

Detection rates showed the expected parametric effect; the higher the number of presented distractors prior to the cue, the higher the resulting distractor-induced deafness effect. Descriptive statistics for detection and error rates are given as an overview in Table 1 and are depicted in Figure 1. The repeated-measures ANOVA revealed a significant effect of the number of distractors with F(2, 42) = 10.862, *p* = 0.002, ω2 = 0.145. In Bonferroni-corrected post hoc analyses, the differences in task performance between the numbers of distractors were compared. Significant differences could be found between zero and three distractors (*p* = 0.012, MDiff = −8.03, 95%-CI [−14.60, −1.45]) and zero and six distractors (*p* < 0.001, MDiff = −12.07, 95%-CI [−18.64, −5.49]). These findings clearly indicate that expression of experimentally induced deafness is modulated by distractors.

A further modulation of the distractor-induced deafness by tDCS, however, could not be found. The tDCS effect was not significant with F(2, 42) = 0.577, *p* = 0.582, ω2 = 0.000 as well as the interaction of numbers of distractors and tDCS with F(4, 84) = 1.443, *p* = 0.238, ω2 = 0.003.

### 3.2. Error Rates for Missed Cues

Error rates for missed cues are almost absent when no distractors were presented. The presentation of distractors leads to only a slight increase in errors, resulting in a not significant effect for the numbers of distractors with F(2, 42) = 1.233, *p* = 0.296, ω2 = 0.004.

However, while mere tDCS showed no significant effect with F(2, 42) = 0.181, *p* = 0.780, ω2 = 0.000, the interaction between tDCS and number of distractors appeared to have an influence with F(4, 84) = 4.532, *p* = 0.006, ω2 = 0.066. Figure 1b shows how the error rates increase depending on the number of distractors and tDCS conditions. Real tDCS led to higher error rates in trials with three distractors and to lower error rates in trials with six distractors compared to sham stimulation. In post hoc analyses, however, no significant differences could be observed.

## 4. Discussion

The purpose of the present study was to investigate whether distractor-induced perceptual impairments occur for clearly distinguishable different auditory stimuli. Moreover, the possible influence of tDCS on detection and error rates was examined. The findings can be summarized as follows:In line with the hypothesis, the presentation of target-like but irrelevant stimuli (distractors) reduces the probability of having conscious access to a target.Detection rates showed to be unaffected by tDCS. Descriptive statistics showed a slight increase in detection rates under stimulation compared to sham stimulation, but this influence was neither significant nor relevant.Error rates for missed cues were affected by tDCS in dependence on the number of distractors, such that error rates decreased with six distractors under stimulation compared to sham stimulation. In contrast, there was an increase in error rates for three distractors under stimulation.

These findings indicate that the effect established for the processing of visual stimuli can be extended to auditory stimuli also with this setting. In all cases, the expression of the perceptual impairment is significantly amplified by the number of distractors sharing the feature of the target. Following the neuro-computational model [8], a similar inhibition mechanism can be assumed. Although the effect of auditory distractors on target detection is comparable with distractor-induced blindness found in vision, it seems to be less pronounced. Usually, detection rates are reported to be reduced to 60–70% (e.g., [2,6,16]) for visual and to about 75% for auditory stimuli [5]. In this study, only a reduction to 83.6% was achieved, but the effect size was nearly identical to those found in other studies using auditory distractors [5,13].

Following the idea that the inhibition process is driven by top-down control, it is surprising that frontal tDCS had no effects on detection rates. On the one hand, tDCS has been applied frequently to neuropsychological tasks that are not sensitive enough to the subtle effects of its stimulation [9]. The results of this study might therefore indicate that distractor-induced deafness is such a task and that other techniques, such as transcranial magnetic stimulation, might be better suited for further research, especially when considering the aforementioned limitations of tDCS, such as strong heterogeneity in possible effects and large inter- and intra-individual variability among test subjects [10,12]. On the other hand, tDCS effects on target detection might have been masked by other, stronger effects, such as the number of distractors. It was pointed out that often, in moderate-sized samples, behavioral effects of tDCS might be too small to differ significantly from sham stimulation [9]. Furthermore, the effect of tDCS seems to interact with task difficulty in terms of stronger effects for tasks with great difficulty [21,22,23]. In the present study, the difficulty of the task was very low; even in the condition with six distractors, the detection rates were above 80%. However, since further increasing the number of distractors leads to saturation effects [5], the number of distractors used in this study leads to the most difficult variant of this paradigm.

Additionally, task familiarization or learning might interfere with tDCS effects. In this study, each subject participated in three sessions, but the order of stimulation and sham conditions was balanced. Consequently, the detection rates did not change systematically over the measurements.

There are also a number of further variables, namely differences in polarity, current intensity, electrode size, and volunteer group, from which some cannot be controlled easily, possibly leading to different modulatory effects [9,24,25,26,27]. Furthermore, studies showed that the spatial resolution of tDCS is too low to precisely stimulate a certain area [9,28]. Consequently, it might be possible that the stimulation in this experiment was too unspecific to the PFC to induce an effect. Using alternative stimulation protocols such as high-definition tDCS, in which multiple small electrodes are used to create a more focal current, alternating current stimulation or random noise stimulation may more effectively modulate distractor-induced effects. Furthermore, the bilateral frontal stimulation used in the present study is problematic if both stimulated frontal areas would contribute equally to the distractor-induced effects. In this case, activating and inhibitory effects would cancel each other in both stimulation conditions.

Although the stimulation had no overall effect on detection rates, its influence on the errors for missed cues seems clear. It is unclear, however, whether participants committing the cue error consciously perceived the cue but forgot or did not see the cue at all. Since the SOA in this experiment always was set to 0 s, it would be possible to let participants react to the cue–target dyad immediately after perceiving the cue and target, instead of reacting after the whole trial, in order to see if the waiting time until the end of a trial makes a difference on this effect. The neuro-computational model [8] identified the decay time constant of activity in input cortices, which was used to mimic the iconic memory, as one factor influencing distractor-induced effects. It could be hypothesized that tDCS affected the iconic memory of participants by prolonging distractor-induced cortical activity, leading to the increase in cue error rate from zero to three distractors. The decrease in cue error rate from three to six distractors could be explained by overlapping cortical activity resulting in not all distractors inducing a response. In order to test this hypothesis, the time between the last distractor and the cue being presented should be varied systematically in further studies. If the effect disappears with longer time intervals between the last distractor and the cue, this may be a hint for the iconic memory being a plausible source of the effect. Another potential source for the cue error rate effect is the possible stimulation of the inferior frontal junction (IFJ), which is responsible for representations of abstract properties of visual objects and consistently active in change of tasks [29]. A low processing capacity of the IFJ was revealed [30], which could lead to a bottleneck effect in information processing. One could argue that in the setup of the distractor-induced deafness paradigm, the task changes from identifying the visual cue to hearing the auditory target. The perception of distractors might lead to confusion about the order of tasks and, therefore, higher performance costs, resulting in an inhibition of stimulus detection. In addition, it was shown that tDCS had a significant effect on task-switching costs [31].

A possible limitation of the present study is the probability of sham guessing. The tDCS conditions have to be blinded properly in order to produce interpretable results. The blinding procedure using the well-established fade-in/out sham protocol in this study has proven to be an effective way of keeping participants unaware of the stimulation conditions they receive [32,33]. However, it has been argued that experiments with a repeated-measures design can be more sensitive to unblinding since participants undergo multiple sessions, which they can compare with respect to sensations they felt during stimulation and with respect to their subjective level of performance [34]. It has been shown that the fade-in/out sham protocol mimics real stimulation successfully and cannot be guessed by participants [34]. In addition, the control condition without a target leaves no way to accurately assess the performance in the distractor-induced deafness, so it can be assumed that the procedure chosen here was unproblematic with respect to sham guessing.

Thus, further research in this area, in addition to varying various parameters such as the current intensity of stimulation, should increase the difficulty of the deafness task. However, since an increased number of distractors leads to saturation effects [5], research should rather vary the tone frequency, the presentation duration of auditory stimuli, or the temporal intervals between distractors. Since stimulation of the frontal cortex seems, in principle, suitable to modulate the effects of distractor-induced deafness, other methods of influencing this brain region, such as TMS or even sleep deprivation, might also be suitable to gain a further understanding of the processes underlying distractor-induced perceptual impairments.

In sum, in this study, the chosen setup led to successful replication of behavioral deafness effects. The results under tDCS stimulation indicate that the chosen paradigm may have difficulty too low to respond to stimulation. That the error rates nevertheless led to a tDCS effect may be due to the divided attention between the visual cue and the auditory target.

## Figures and Tables

**Figure 1 brainsci-12-00738-f001:**
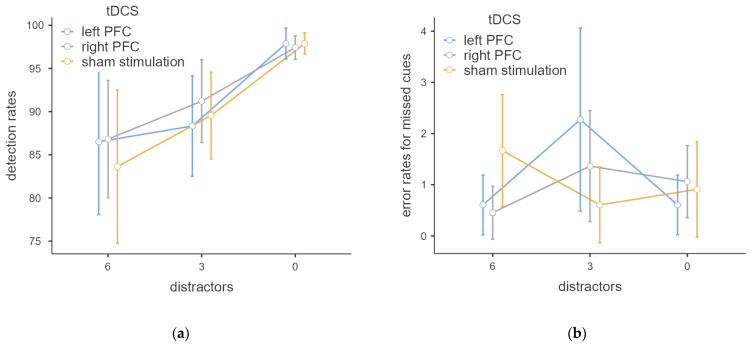
Detection and error rates for missed cues (given as percentages), error bars indicate 95% confidence intervals. (**a**) Detection rates depending on number of distractors and stimulation condition; (**b**) error rates for missed cues depending on number of distractors and stimulation condition.

**Table 1 brainsci-12-00738-t001:** Detection rates and error rates for missed cues depending on number of distractors and stimulation conditions (given as percentages, 95% confidence intervals in parentheses).

	tDCS	6 Distractors	3 Distractors	0 Distractors
Detection rates	left PFC	86.5 (78.1, 95.0)	88.3 (82.5, 94.2)	97.9 (96.1, 99.7)
	right PFC	86.8 (80.0, 93.6)	91.2 (86.4, 96.0)	97.4 (96.1, 98.8)
	sham stimulation	83.6 (74.8, 92.5)	89.5 (84.5, 94.6)	97.9 (96.6, 99.1)
Error rates for missed cues	left PFC	0.61 (0.02, 1.19)	2.27 (0.48, 4.06)	0.61 (0.02, 1.19)
	right PFC	0.46 (−0.06, 0.97)	1.36 (0.28, 2.45)	1.06 (0.36, 1.77)
	sham stimulation	1.67 (0.57)	0.61 (−0.13, 1.35)	0.91 (−0.02, 1.84)

## Data Availability

Data used in this study are available for re-analysis and validation upon request to the corresponding author.

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
