# Peer review of "The Effect of Transcranial Direct Current Stimulation on Error Rates in the Distractor-Induced Deafness Paradigm"

_brainsci, 2022, doi:10.3390/brainsci12060738_

Round 1

Reviewer 1 Report

In this study by Michael and colleagues, the authors applied tDCS over frontal areas of the cortex to investigate whether stimulation alters the perception of distractor information. The authors found that tDCS had no effect on target detection, but appears to have some influence on error rates, as stimulation was increased the rate in specific conditions.  Below are some major concerns:

“Transcranial direct current stimulation (tDCS) is a well-established method to investigate and alter brain physiology [9]. When the anode is placed over a certain area, it depolarizes neuron membranes of that region, which in turn need less additional activity to induce an action potential and are therefore easier to excite, whereas cathodes lead to hyperpolarized membrane, which are harder to excite [10].” Theoretically speaking, this is correct, however the effects of tDCS on both behavior and cortical excitability of humans are well-known to have large inter- and intra-individual variability. This should be considered in both the introduction and as a potential limitation as to why a minimal effect was found in this study.

Along these lines, the authors decided to select a study design in which the anode and cathode were flipped over bilateral frontal regions (e.g. F7 and F8), thus the same two frontal areas were stimulated in the “real” stimulation conditions. The problem with this sort of design is that the authors did not hypothesize whether the left-, right-, or bilateral prefrontal areas would have a role in suppressing/gating irrelevant information. For example, if both sides of the prefrontal cortex are involved and the anode/cathode were to “excite/inhibit” the select brain areas, then would expect stimulation to have no effect on behavior (unless of course either the left or right prefrontal cortex is expected to have a stronger contribution to this process). Thus a few questions arise from this design: Why did the authors not select to specifically stimulate one side of the prefrontal cortex? Why not include another (control) brain region to be stimulated, which might not be involved in this specific cognitive process (e.g. motor cortical areas, or perhaps the cerebellum)?

Further to maximize stimulation effects, I find it curious as to why the others selected 1mA as the current intensity. Would not 2 mA have been more optimal to induce effects on behavior? (of course, individualizing tDCS intensity would be even better).

In terms of the behavior, since one of the main conclusions is that “the results under tDCS stimulation indicate that the chosen paradigm may have a difficulty too low to respond to stimulation”, then why not create a situation in which the baseline is much more difficult for participants to complete (e.g. could not more distractors be added to the design?).  Moreover, as each subject completed three different sessions of the task, could the effect of task familiarization potentially interfere with the effects of tDCS? Finally, I think it is important for the authors to explicitly state the difference of detection rate and error rate in the methods section, as these are the main outcome measures of the task.

“Error rates for missed cues were affected by tDCS in dependence on the number of distractors, such that error rates increased with three distractors under stimulation and decreased with six distractors.” Is this a valid statement to make based on both stimulation conditions? Rather, it seems that stimulation appears (more strongly) to reduce error rates in the 6-distractor task version when compared to sham stimulation.

Author Response

Response to Reviewer 1

We thank the reviewer for his/her overall positive evaluation of our manuscript and for his/her thoughtful  and constructive comments. We believe the manuscript has improved significantly as a result of having addressed the suggestions. Please find our point-by-point responses below.

  1. “Transcranial direct current stimulation (tDCS) is a well-established method to investigate and alter brain physiology [9]. When the anode is placed over a certain area, it depolarizes neuron membranes of that region, which in turn need less additional activity to induce an action potential and are therefore easier to excite, whereas cathodes lead to hyperpolarized membrane, which are harder to excite [10].” Theoretically speaking, this is correct, however the effects of tDCS on both behavior and cortical excitability of humans are well-known to have large inter- and intra-individual variability. This should be considered in both the introduction and as a potential limitation as to why a minimal effect was found in this study.

We agree with you and added this information to the introduction and discussion section.

  1. Along these lines, the authors decided to select a study design in which the anode and cathode were flipped over bilateral frontal regions (e.g. F7 and F8), thus the same two frontal areas were stimulated in the “real” stimulation conditions. The problem with this sort of design is that the authors did not hypothesize whether the left-, right-, or bilateral prefrontal areas would have a role in suppressing/gating irrelevant information. For example, if both sides of the prefrontal cortex are involved and the anode/cathode were to “excite/inhibit” the select brain areas, then would expect stimulation to have no effect on behavior (unless of course either the left or right prefrontal cortex is expected to have a stronger contribution to this process). Thus a few questions arise from this design: Why did the authors not select to specifically stimulate one side of the prefrontal cortex? Why not include another (control) brain region to be stimulated, which might not be involved in this specific cognitive process (e.g. motor cortical areas, or perhaps the cerebellum)?

You have raised an important question; We chose this bilateral bipolar-balanced montage following Khalil et al., 2020 and Neuling et al. 2012.  This montage is well suited to stimulate frontal brain areas without too much involvement of other cortical regions. We preferred this montage primarily to keep more posterior areas out of stimulation, since the neuro.computational model by Ebner et al.  also includes the basal ganglia, thalamus, and motor areas. We added this information tot he methods section.

We agree that this montage leads to a severe limitation given the absent stimulation effect. We adressed this in the discussion section.

  1. Further to maximize stimulation effects, I find it curious as to why the others selected 1mA as the current intensity. Would not 2 mA have been more optimal to induce effects on behavior? (of course, individualizing tDCS intensity would be even better).

We have justified our decision in the methods section: There is evidence that in arrangements requiring different cognitive load, there are no differences between the effects of stimulation at 1mA and 2mA []. This appears comparable to the requirements in the present study, where different numbers of distractors must be ignored in different conditions.

Given the results, alternative stimulation protocols are discussed: High-definition tDCS, in which multiple small electrodes are used to create more focal current, alternating current stimulation or random noise stimulation may more effectively modulate distractor-induced effects.

  1. In terms of the behavior, since one of the main conclusions is that “the results under tDCS stimulation indicate that the chosen paradigm may have a difficulty too low to respond to stimulation”, then why not create a situation in which the baseline is much more difficult for participants to complete (e.g. could not more distractors be added to the design?).  

Since further increasing the number of distractors leads to saturation effects [5], the number of distractors used in this study leads to the most difficult variant of this paradigm. We added this information in the corresponding paragraph.

  1. Moreover, as each subject completed three different sessions of the task, could the effect of task familiarization potentially interfere with the effects of tDCS? 

The order of stimulation and sham conditions was balanced. Consequently, the detection rates did not change systematically over the measurements. We included a new paragraph in the discussion.

  1. Finally, I think it is important for the authors to explicitly state the difference of detection rate and error rate in the methods section, as these are the main outcome measures of the task.

Indeed, we could have been more specific: A more detailed explanation was added to the methods section

  1. “Error rates for missed cues were affected by tDCS in dependence on the number of distractors, such that error rates increased with three distractors under stimulation and decreased with six distractors.” Is this a valid statement to make based on both stimulation conditions? Rather, it seems that stimulation appears (more strongly) to reduce error rates in the 6-distractor task version when compared to sham stimulation.

We think both formulations say the same thing. We have taken up your suggestion to put the condition with 6 distractors  more in the focus in order to present the results more clearly.

Reviewer 2 Report

  • More specific and concrete hypotheses could be provided in the introduction, also emphasizing the originality of the study advancing existing literature.
  • It is written that "A minimum sample size of 19 participants was calculated in advance using G*Power (Faul et al., 2009)." This is a rather small N. Also considering the variable sample with a wide age range. What input was exactly used to obtain this number?
  • "Four additional participants had to be excluded from further analyses due to high error rates in control conditions." Please specify the used criterion for establishing high error rates.
  • Later, it is written that "First, error rates in the control condition were computed. Participants with a rate of more than 30% false alarms in at least one of the control conditions where no targets or no cues were presented were excluded from further analyses. This approach is based on previous studies on distractor induced blindness, where these errors were seen as indicators for lack of task understanding, lack of attention, or random response behavior." Please cite all the considered and necessary studies here.
  • Partial eta-squared is a biased effect size measure. It would be more appropriate and useful to use omega squared.
  • I suggest to report 95% confidence intervals instead of standard deviations or standard errors in the graphs and tables. These are more meaningful and useful (also for meta-analyses).

Author Response

Response to Reviewer 2 

We thank the reviewer for his/her overall positive evaluation of our manuscript and for his/her thoughtful  and constructive comments. We believe the manuscript has improved significantly as a result of having addressed the suggestions. Please find our point-by-point responses below.

  1. More specific and concrete hypotheses could be provided in the introduction, also emphasizing the originality of the study advancing existing literature.

We agree with you and have incorporated this suggestion throughout our paper.We have revised the last paragraph of the introduction to make the hypotheses and their rationale clearer.

  1. It is written that "A minimum sample size of 19 participants was calculated in advance using G*Power (Faul et al., 2009)." This is a rather small N. Also considering the variable sample with a wide age range. What input was exactly used to obtain this number?

Indeed, we could have been more specific: Calculation was conducted for f = 0.25, α = .01; 1-β = 0.80, we added this information in the respective section. We had no exclusion criteria regarding the age of our subjects; the age range was due to opportunity sampling.

  1. "Four additional participants had to be excluded from further analyses due to high error rates in control conditions." Please specify the used criterion for establishing high error rates.

Done.

  1. Later, it is written that "First, error rates in the control condition were computed. Participants with a rate of more than 30% false alarms in at least one of the control conditions where no targets or no cues were presented were excluded from further analyses. This approach is based on previous studies on distractor induced blindness, where these errors were seen as indicators for lack of task understanding, lack of attention, or random response behavior." Please cite all the considered and necessary studies here.

Done.

  1. Partial eta-squared is a biased effect size measure. It would be more appropriate and useful to use omega squared.

Done.

  1. I suggest to report 95% confidence intervals instead of standard deviations or standard errors in the graphs and tables. These are more meaningful and useful (also for meta-analyses).

Done.